# The Androgen Regulation of Matrix Metalloproteases in Prostate Cancer and Its Related Tumor Microenvironment

Carmela Sorrentino [1,†], Rosa D'Angiolo [1,†], Giulia Gentile [1], Pia Giovannelli [1], Bruno Perillo [2], Antimo Migliaccio [1], Gabriella Castoria [1] and Marzia Di Donato [1,*]

1 Department of Precision Medicine, University of Campania 'L. Vanvitelli', Via L. De Crecchio, 80138 Naples, Italy
2 Istituto di Scienze dell'Alimentazione, Consiglio Nazionale delle Ricerche (C.N.R.) 83100 Avellino, Italy
* Correspondence: marzia.didonato@unicampania.it
† These authors equally contributed to this work.

**Abstract:** Prostate cancer represents the most common type of cancer among males and the second leading cause of cancer death in men in Western society. In most cases (~70%), PC has a slow and symptom-free growth, whereas it is more aggressive in the remaining patients. Current PC therapies prevalently target the proliferative function of the androgen receptor and may only be effective within short periods, beyond which the disease will progress to metastatic and castration-resistant phenotype. Preclinical and clinical studies are aimed at investigating the molecular basis for prostate cancer spreading. Although considerable efforts have been made to dissect the programs that foster prostate cancer spreading, few biomarkers predictive of metastatic phenotype have yet been identified and few therapeutic options are available for treatment of the metastatic disease. In the present paper, we will discuss innovative aspects of prostate cancer biology, which impinge on the role of cancer-associated fibroblasts and the released matrix metalloproteinases in the disease progression. Investigating these aspects might allow the discovery of clinically actionable biomarkers to target in the advanced stages of prostate cancer.

**Keywords:** androgens; androgen receptor; metalloproteases; prostate cancer; tumor microenvironment; carcinoma-associated fibroblasts; extracellular matrix; invasiveness

## 1. Introduction

The tumor microenvironment (TME) plays a fundamental role in cancer development and progression [1]. Cancer cells recruit adjacent cells to collaborate in survival strategies. TME represents an intricate network constituted of several cell types, such as immune and endothelial cells, fibroblasts, myofibroblasts, and adipocytes. In addition, various soluble factors, such as growth factors, components of the extracellular matrix (ECM), including collagen, hyaluronic acid, and elastin, cooperate each other to drive tumor progression [2,3]. The complexity of TME remains, however, the subject of numerous studies aimed at understanding how it promotes cancer progression [2,4]. As such, stromal cells represent an important and attractive target for new drugs. Cancer-associated fibroblasts (CAFs) are the most abundant cellular playmakers in TME [5]. They regulate tumor invasion by producing proinvasive stimuli [6]. Additionally, through cell–cell contacts, secretion of ECM components, growth factors, and chemokines, CAFs might support angiogenesis, tumor progression, drug resistance, and immune-escape. The most significant soluble factors secreted by CAFs include transforming growth factor (TGF-β1), insulin-like growth factor 1 (IGF-1), hepatocyte growth factor (HGF), platelet-derived growth factor (PDGF), interleukin-6 (IL-6), and fibroblast growth factors (FGFs). Notably, matrix metalloproteinases (MMPs) are also released by several cell types of TME [4].

MMPs enzymes belong to the zinc endopeptidases family for the presence of $Zn^{2+}$ in the catalytic site. They are secreted or anchored to the cell basement membrane.

TME and MMPs play a crucial role in the progression of prostate cancer (PC) [7–9], and very recent studies have highlighted the importance of the androgen/androgen receptor (AR) axis in controlling the functions of CAFs in ex vivo cultures from PC patients [10,11]. Among the responses elicited by androgens in prostate CAFs, the release of MMP-2 is worth noting [10]. These and other previous findings in the literature [12–14] point to the critical role of MMPs in PC pathogenesis and progression.

In this study, we will revise recent findings on the role of MMPs in PC progression, focusing on prostate TME and emphasizing the link between MMP's release and androgens.

## 2. PC: Facts and Perspectives

The International Agency for Cancer Research recently reported that PC is the second occurring cancer in men and the most frequently diagnosed in the world [15]. It represents the fifth-leading cause of death in men, with an incidence rate ranging from 6.3 (in Asia and North Africa) to 83.4 (in Northern and Western Europe, Australia, North America, and South Africa) in 100,000 men. The mortality rate is less variable than the incidence. In fact, PC is the leading cause of death in areas such as Sub-Saharan Africa, the Caribbean, Central and South America, and Sweden [16]. In Italy, PC still represents the most common cancer in men, accounting for 18.5% of all diagnosed cancers. Over the last decade, a high incidence has been reported (36,074 new cases/year). Nevertheless, the mortality rate has decreased by −16.6%, with a good life expectancy five years after diagnosis, also considering the advanced average age of patients [15,16]. Five-year survival in the PC in all its stages has increased by 98% and the survival rate is lower for black individuals, as compared with white individuals [15].

Prostatic tissue development and neoplastic transformation are related to steroid hormones produced by male gonads, mainly the androgens. AR remains one of the principal targets in PC therapy. Androgen deprivation therapy (ADT), the first-line approach in PC, aims to deplete androgens to contrast PC growth and progression [17,18]. Novel AR pathway inhibitors have successfully entered the routine management of PC patients [19–21]. Nonetheless, these therapies frequently fail, and PC becomes androgen-independent, thus acquiring ADT resistance. At this stage, castration-resistant PC (CRPC) may eventually spread and metastasize [19]. Recent studies have demonstrated that several mechanisms are involved in ADT resistance of PC. They include AR mutations [22], amplification [23] and alternative splicing [24]. Other mechanisms, related to aberrations and derangement of signaling pathways, deregulation of AR coactivators, and aberrant expression of tyrosine kinase receptors (RTKs), however, might be also involved ([25] and references therein). Nevertheless, these findings still leave open the question of drug response in PC.

Nowadays, different therapeutic approaches aim to decrease the androgen levels or block the receptor functions. As such, inhibition of CYP17A1 by abiraterone [26], use of second-generation AR antagonists, including enzalutamide, apalutamide, and darolutamide [27–29], addition of docetaxel in combo therapies [30], use of PARP inhibitors [31], or bipolar androgen therapy (BAT) [32] have successfully entered the clinical management of PC patients. Other therapeutic options might be, however, actionable. They include radical prostatectomy, external beam radiotherapy (RT), brachytherapy and cryotherapy [33], use of new tracers for positron emission tomography/computed tomography (PET/TC), radiopharmaceuticals such as radium-223, and novel focal therapies, which have provided important advances in diagnosis, follow-up, and treatment in advanced stage of metastatic PC disease [34]. Nonetheless, the mortality rate in CRPC patients with metastatic disease remains elevated.

*The Role of Stroma in PC Development and Carcinogenesis*

The human prostate is not only composed of epithelial cells but also of fibromuscular connective tissue [35] and mesenchymal cells. The prostate gland originates from the urogenital sinus formed by epithelial (UGE) and mesenchymal (UGM) cells, both necessary for prostate development. In vivo experiments have shown, indeed, that whether

UGE and UGM are grafted separately into nude mice, neither of the two compartments can correctly differentiate [36,37]. However, the normal prostate epithelial compartment, composed of ducts and exocrine glands, interacts with the surrounding fibromuscular connective tissue stroma [35]. This interaction guarantees the correct organogenesis and the maintenance of normal prostate function and maturity [38]. In the process of prostate organogenesis, the androgens take part, mainly testosterone [39], but so too do the "andromedins" [40]. In particular, the ligand-activated stromal AR triggers the production and release of these paracrine factors, or andromedins, by mesenchymal cells themselves. Andromedins, in turn, regulate prostate epithelial growth and differentiation [41,42]. The more studied andromedins are the FGF7, FGF10, and the PDGF [43]. FGFs are important in the prostate gland, as relevant levels of FGF2, FGF7, and FGF9 can be detected in normal prostate stromal cells, while the cognate receptors, FGFRs, are expressed in secretory prostatic epithelium. The FGF/FGFR signaling, hence, becomes involved in the control and homeostasis of the normal prostate gland. Derangement of this axis promotes different pathologic conditions, including the prostatic intraepithelial neoplasia (PIN), the carcinoma "in situ", and, lastly, the invasive and metastatic PC [44]. Again, PDGF family members have been found in the PC stromal compartment where they can promote cell migration, proliferation, and transformation [45].

Throughout life, ageing can promote molecular and structural changes in TME, accounting for many pathological processes, including benign prostate hyperplasia, prostatitis, and, lastly, PC [46,47]. In this process, many stromal cells, embedded in the ECM, interact with neoplastic epithelial cells, taking part in the prostate tumor [9,48]. The tumor-associated stromal cells which promote cancer development and progression are represented by myofibroblasts, adipocytes, smooth muscle cells, lymphocytes, endothelial cells, pericytes, macrophages, and mast cells. Epithelial cancer cells, in turn, sculpt the microenvironment through the secretion of various cytokines, chemokines, and other factors [8]. Thus, a dialogue among the different compartments is established and this feature fosters tumor growth, metabolic rewiring, stemness, and metastatic events [8,48]. In this context, the role of CAFs in PC progression [9,49], is undeniable. CAFs exhibit significant expression levels of the wild-type AR, irrespective of Gleason's score [10]. The AR acts through nongenomic effects [10] but also influences CAF-specific AR-driven transcriptional programs (e.g., regulation of promigratory cytokines release [50]). Thus, CAFs might play a role in the PC transition towards CRPC after ADT and influence the response to ADT by promoting PC progression towards neuroendocrine (NEPC) differentiation. This process might be due to an expansion of a specific subpopulation of CAFs that induce NEPC transformation through Wnt-SFRP1 signaling [51], or epigenetic modifications [52]. In the neoplastic transformation of the prostate gland, the andromedins can also take part. Particularly, PDGF-D and PRGFR-β are highly detectable in the PC stromal compartment as well as in primary and metastatic lesions [53,54]. Their expression is directly related to the recurrence of the disease, [53] thus suggesting a prognostic role for PDGFR-β in PC [53]. However, further studies are needed for a deep understanding of the liaison between CAFs and PC cells.

In addition to CAFs, T-lymphocytes, B-lymphocytes, tumor-associated macrophages (TAMs), tumor-associated neutrophils (TANs), and dendritic cells (DC) also express AR, exhibiting different roles [55]. PC cells are surrounded by CD8+ and CD4+ subsets of T-lymphocytes. The AR expressed by these cells is mainly involved in the production of cytokines [56,57], which affect PC development and progression [58]; however, the molecular mechanism is yet to be described. Similarly, the effect of androgens on innate immune cells functions is unexplored. In the inflammatory microenvironment, the role of TAN is almost indistinct and complicated. As occurs for macrophages, TAN can also display protumoral and antitumoral capabilities [59] and, thus, can be classified as N1-like (antitumoral neutrophils) or N2-like (protumoral neutrophils). The full range of mechanisms responsible for the pro- vs. antitumor effects of TANs has not yet been elucidated [49,59]. However, the ability to identify the different neutrophil subpopulations

in the tumor is critical to understand TANs evolution and contribution throughout tumor progression. Findings have shown that AR overexpression mediates the expansion of human neutrophils population, while its knockdown impairs neutrophils proliferation. Thus, it is still to be elucidated whether AR supports the N1- or N2- TAN phenotype [50]. It is noteworthy that a high neutrophils/lymphocytes ratio seems to be associated with resistance to abiraterone and docetaxel treatment in metastatic CRPC patients [60–62]. In vivo studies have shown that AR knockdown reduces neutrophil proliferation and maturation, through the inhibition of STAT3 and ERK activation, and the reduction of the production of chemokines and cytokines such as IL1-β, IL-6, and TNF-α [63]. Again, TAN might trigger the cancer cells proliferation and invasiveness and the new vessels formation through the release of MMP9 and VEGF.

Future investigations in this direction might allow the development of precision strategies for PC therapy. Most current therapeutic options in PC patient management are, indeed, based on drugs that affect epithelial cancerous cells. The simultaneous targeting of TME and epithelial cells might increase the likelihood of favorable patient outcomes, even when the AR-based therapies are unsatisfactory. In the next sections, we will present and discuss the more recent findings on the cross-talk between CAFs and PC cells, with a particular focus on the released molecules, such as MMPs.

## 3. The Role of CAFs in ECM Remodeling

Cancer cells and the surrounding TME cannot be considered as two separate environments. There is, indeed, a dynamic and reciprocal relationship between them, as shown in Figure 1 [64].

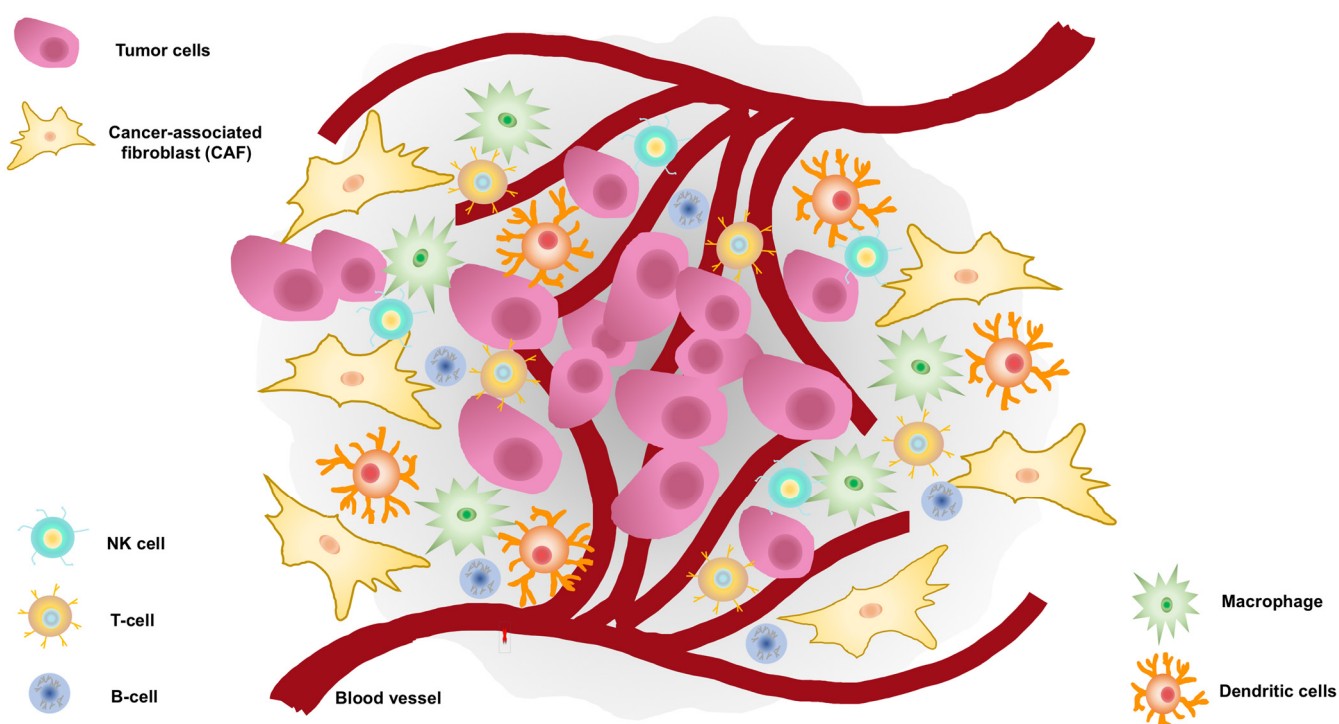

**Figure 1.** Tumor microenvironment (TME) surrounds prostate cancer cells.

TME encompasses cellular and noncellular components. The TME cellular component surrounds tumor cells and consists of endothelial cells, carcinoma-associated fibroblasts (CAFs), lymphocytes such as B-cells, T-cells, and natural killer (NK) cells, tumor-associated macrophages, and dendritic cells. The noncellular component is represented by the extracellular matrix which acts as a scaffold.

CAFs are the major component of stroma and play a role in the crosstalk with PC cells [65]. However, molecular characterization of CAFs has not yet been defined, as they are more frequently described as a "cellular state". Recent studies indicate that more than one cell type can be recruited as CAFs precursors. Such cells include resident fibroblasts, peritumoral adipocytes, bone marrow mesenchymal stem cells, hematopoietic stem cells, epithelial cells (that underwent epithelial-to-mesenchymal transition EMT), and endothelial cells (after endothelial-to-mesenchymal transition: EndMT) [66–68]. A subgroup of precursors can be recruited and activated by complex processes to acquire the CAF status. Epigenetic modifications, changes in noncoding RNA expression, and abnormal activation of some signaling circuits, such as NFκB, IL-6/STAT3, FGF-2/FGFR1, and TGF-β/SMAD, can be involved in CAF origin [68–70]. CAFs typically exhibit some markers, including α-smooth muscle actin (α-SMA), fibroblast activation protein (FAP), and platelet-derived growth factor receptor (PDGFR)-α and β, which are similar to myofibroblast's markers [69]. Expression of these specific molecules helps their isolation in vitro and is currently used for new specific therapeutic applications [71,72].

Prostate CAFs promote the normal gland transformation, the transition of the benign prostate hyperplasia-derived cells versus a malignant phenotype and PC progression [73]. Once "activated", they have different roles in ECM remodeling, including deposition, modification, degradation, and organization. Such effects are due to their ability to produce ECM components and degrade them by releasing MMPs. CAFs, indeed, produce different collagen types, such as type-I and -III, which increase matrix stiffness and promote PC cell migration, invasion, and spreading [74]. Moreover, they deposit fibronectin by triggering the formation of an oriented fiber's network that, through the integrin-αv, favors CAFs–PC crosstalk and migration of cancer cells [75]. Again, CAFs promote the proliferation and invasion of the epithelial counterpart in vitro [73] and trigger the formation of distant metastases in mouse models [76]. Relevant to PC progression, CAFs even induce the ADT [77] and constitute a "niche" that sustains the function of cancer stem cells (CSCs), thereby promoting the metabolic reprogramming of PC cells or their epithelial–mesenchyme transition (EMT), which often characterize the metastatic spreading [78,79].

On the other hand, epithelial PC cells modify TME by releasing cytokines, chemokines, growth, and survival factors, as well as hormones [9,80]. Noteworthy, PC might by itself produce the androgens, which act locally to self-sustain the growth of epithelial transformed cells, and likely of surrounding cells, provided they express AR. This game played by cancer and the surrounding stromal cells fosters tumor growth, metastasis, and therapy escape.

The latter findings raise the question of the androgen responsiveness as well as AR expression of CAFs. These cells express low, but significant, levels of AR [10,81], which drive the prostatic epithelial malignant transformation [82]. AR loss is related to low levels of fibroblast-released growth factors (e.g., FGF-2 and FGF-10) and reduced potential of PC carcinogenesis [82]. The stromal AR somatic knockdown impairs the proliferative and invasive potential of cocultured PC3 cells, suggesting a role for stromal AR in sustaining the growth and invasion of the androgen-independent PC cells [81]. Again, the ligand-bound AR promotes the migration of CAFs and other mesenchymal cells (such as normal fibroblasts and human fibrosarcoma cells, HT1080) [10,83] through the assembly of the AR/FlnA complex and the consequent activation of Rac1 and focal adhesion kinase (FAK) [10,84]. The AR somatic knockdown or its inhibition by specific antagonists in myofibroblast stromal cells WPMY-1 or in CAFs reduce PC cell invasion [85], PC growth [11], and PC spheroids size [10] in different coculture approaches in vitro and in vivo.

By contrast, AR knockdown downregulates the adhesion proteins and upregulates ECM-degrading enzymes [86,87] and stemness markers in PC cocultures [11]. Consistent with the protective role of AR, it has been reported that the receptor activation delays PC cell growth [88], further corroborating the idea that AR blockade might paradoxically foster PC progression. Nevertheless, different studies support an oncogenic role for stromal AR. The conflicting results so far obtained might be related to the different experimental conditions, the approaches used, and the purity of the CAFs population [82,89].

In summary, the findings so far presented point to the role of CAFs in PC. The questions concerning "how" ones move towards the others and "how" they reshape the ECM will be discussed in the subsequent paragraphs.

## 4. MMPs in ECM Remodeling and PC Metastatic Progression: The Androgens Contribution

ECM represents a barrier to cancer cells spreading [90]. The process of spreading and metastasis through the lymphatic system or blood vessels to distant sites is a complex cascade of events that includes the local invasion, intravasation, survival in the circulation, formation of microniches in distant organs, extravasation, adaptation in a foreign microenvironment, and micrometastasis formation. At last, once supported by angiogenesis, the formation of a secondary tumor occurs [91,92]. MMPs play a crucial role throughout all these steps [93,94]. They are proteolytic enzymes belonging to the metzincin superfamily, a large group of proteases characterized by a zinc-ion-binding methionine turn sequence within the catalytic domain [95,96]. This enzyme superfamily can be divided into subgroups based on structural and functional features. The matrixin family is composed of the soluble MMPs and the membrane-type MMPs (MT-MMPs), which are physiologically regulated by the tissue inhibitor of MMPs (TIMP) family [97,98]. These enzymes add specificity to the target for degradation [99]. MMPs are regulated at multiple levels. As shown in Figure 2, they are synthesized as pre-proMMPs, from which the signal peptide is removed during translation to generate proMMPs.

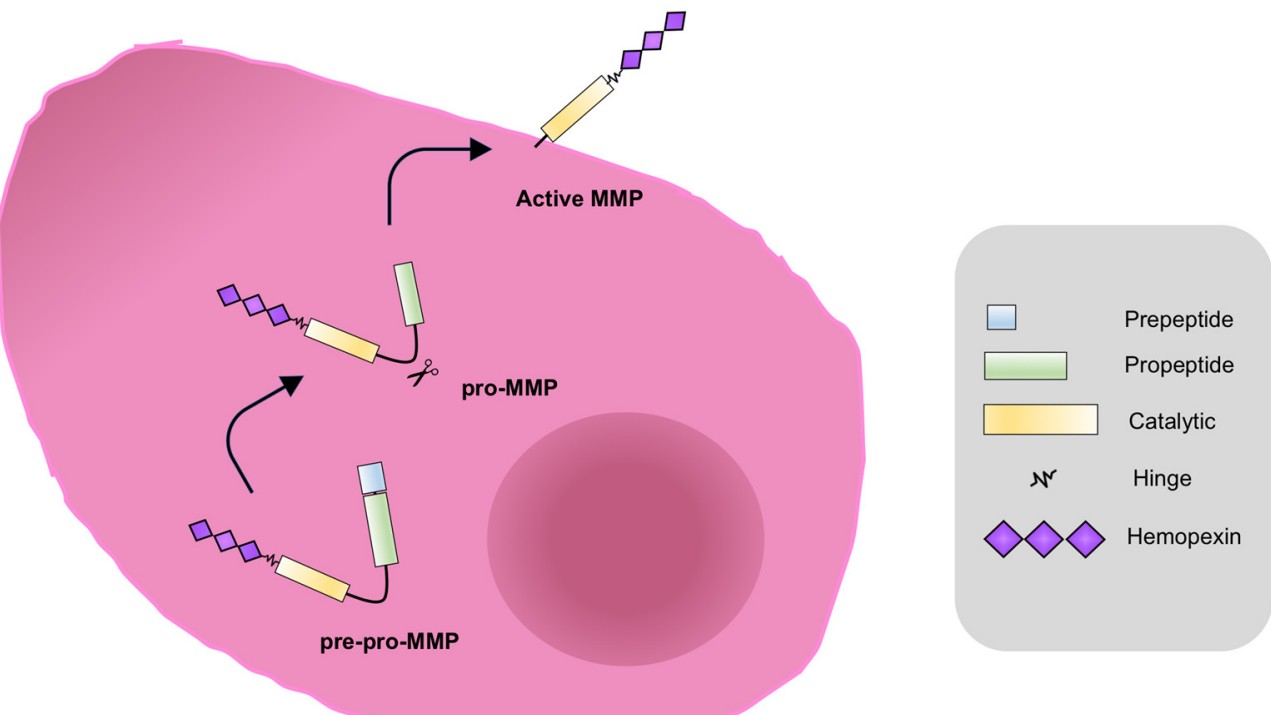

**Figure 2.** MMPs activation. MMPs are synthesized as pre-proenzymes (pre-pro-MMPs), with the amino-terminal signal peptide which drives the enzyme to the endoplasmic reticulum. The signal peptide (light blue) is removed during translation to generate proMMPs. In this state, enzymes are catalytically inactive for the interaction between the thiolic group (SH) of a cysteine residue of the pro-domain with the $Zn^{2+}$ of the catalytic site. In order to process the proMMPs and produce the active MMP form, the propeptide domain (green) cleave occurs. This event is catalyzed by other proteolytic enzymes such as serine proteases, the endopeptidase furin, plasmin, or other MMPs, as well as the membrane-type protein MT-MMP [100].

Once activated, MMPs can modulate the global proteolytic potential in the extracellular milieu.

The adamalysin family consists of disintegrin and metalloproteinases (ADAMs) as well as disintegrin and metalloproteinase with thrombospondin motifs (ADAMTS) sub-groups [101]. Over the years, however, MMPs have been found to regulate pathways involved in apoptosis, immunity, cellular migration, wound healing, and tissue repair, as well as angiogenesis [102]. It is well accepted that MMPs have a key role in PC spreading, given their ability to remodel the ECM. Therefore, they might be considered as a potential source of biomarkers and/or targets for therapeutic interventions. MMPs, indeed, can be found in serum from PC patients. Particularly, MMP-2, -3, -7, -9, -13, -14, -15, and -26 are related with advanced or metastatic disease, while MMP-1 is associated with lower grade tumors [14]. Noticeably, as shown in Figure 3, PC and stromal cells both release different MMPs, such as MMP-1, MMP-2, and MMP-9, and express high levels of MT-MMP1, ADAM-9, -10, -11, -15, and -17, as well as low levels of TIMPs [12,103,104], and the imbalance between MMPs and TIMPs allows PC cell invasiveness [14,105].

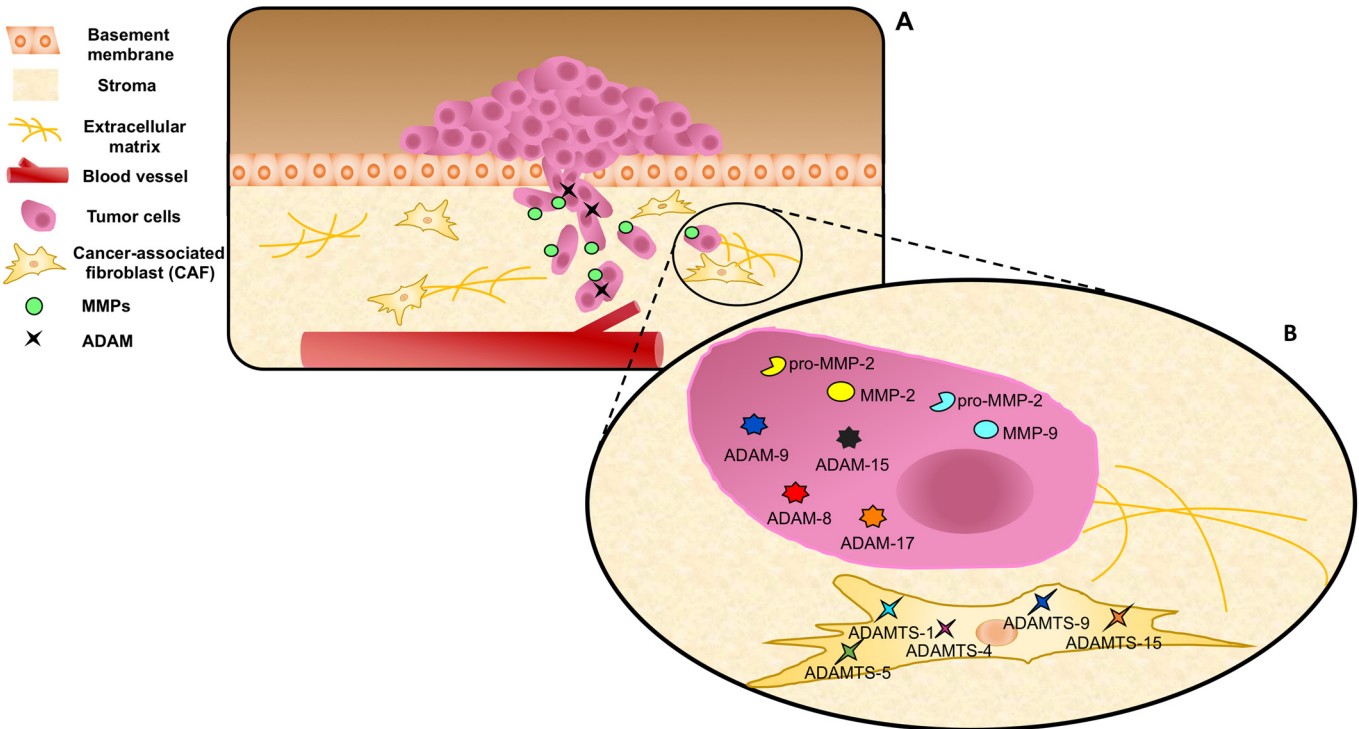

**Figure 3.** MMPs, ADAMs, and ADAMTs expression and release in prostate cancer and stromal cells. (**A**) Cancer cells pass through the MB invading the stroma. MMPs and ADAMs are expressed and/or released by epithelial cancer cells. (**B**) ADAMTs are expressed by CAFs, representing a marker of reactive stroma. Dynamic interactions of tumor cells, CAFs, MMPs, and ADAMs/ADAMTs are represented during ECM remodeling.

Several findings in the literature have shown that CAFs shape the ECM, thus enabling cancer cells growth and spreading, mainly through three interconnected mechanisms: firstly, the production of ECM molecules, then the secretion of ECM-remodeling MMPs, and, lastly, the biomechanical and topographical modifications of ECM fibers [106]. Many reports have shown that accumulation of ECM molecules, including hyaluronan and collagen, in PC stroma is related to several malignant features, such as high Gleason's score and poor clinical outcome in PC [106–108]. Again, CAFs remodel the fibronectin network. In this way, they promote the formation of CAF/PC cell units able to comigrate [75]. Genetic knockdown of MMP-2, -7, or -9 impairs the tumor vascularity and metastasis formation in mouse models. Remarkably, many findings have supported the idea that the tumor–mesenchyme crosstalk is a fundamental step for the complete activity of MMPs [109]. Again, many findings have pointed to the role of ADAM and ADAMTS families in PC progression. ADAM-9, -8, and -15

are higher expressed in malignant, as compared with benign, prostate tissues. Additionally, their expression is significantly associated with higher cancer stages, including positive nodal status, and higher Gleason's scores [110–112]. ADAMTS-1, -4, -5, -9, and -15 seem to represent specific markers of PC reactive stroma [113]. Given these findings, MMPs inhibitors (MPIs) have been employed in preclinical models of human cancers in the past. Nevertheless, these therapies have failed [9], although new MMP-targeted therapies are currently under consideration for their promising antitumor potential. Some of them are being translated into early clinical trials (NCT00695851; NCT00003343) [114].

Diagnostic and therapeutic strategies that impinge on CAF signaling pathways and/or effector molecules have emerged as an important tool for different solid tumors, including PC [69]. Despite the recent advancements in this field, therapies that specifically target the functions of stromal cells are still unavailable and they remain the subject of intense and in-progress mechanistic and functional investigations. In this continuously evolving scenario, different clinical studies (NCT00459407; NCT05311618) are noteworthy, as they aim to associate the decrease of MMP-2 and MMP-9 expression levels with a favorable therapy-response in PC or CRPC patients. As such, MMP-2 and MMP-9 might emerge as new biomarkers of prognosis in PC. It remains, however, to be understood how and whether the androgens regulate the release and activation of MMPs in PC. It is quite certain that AR/androgens regulate MMPs expression and release and not vice versa. AR triggers molecular mechanism culminating with cancer cell spreading and infiltration in different types of cancer [10,115], and in this context, MMPs, particularly MMP-2 and MMP-9, play important roles in the TME to increase invasion and metastasis [116]. Thus, in addition to representing evaluable predictive markers, AR and MMPs may be targeted by specific AR inhibitors and MPIs. The rational for these combinatorial approaches are the distinct molecules, but also complementary pathways, which are targeted and therefore might result in synergistic effects.

The functional link between AR and the key proteins controlling PC metastasis, such as MMPs, remains largely unknown. Several findings have shown that androgens modulate MMPs, ADAMs, and ADAMTs gene expression and activity in PC and its surrounding TME [117–119]. Elevated levels of MMP-9 mRNA in biopsies from PC patients correlate with low disease-free survival [120]. PC metastatic lesions express higher levels of AR and MMP-9 if compared to primary tumors, suggesting a link between the expression of these proteins and the higher risk to occur in a more aggressive PC phenotype [120]. Androgens positively control the expression and activity of MMP-2 or MMP-9 in PC and CRPC-derived cells [118–120], and AR overexpression promotes the activation and release of MMP-9 in VCaP and PC3 cells. The AR overexpression is also related to an increased expression of the phosphatidylinositol 4-phosphate 5-kinase type-1 alpha (PIP5K1$\alpha$), an intermediate of the PI3K/AKT pathway [120]. Thus, the AR-induced MMP-9 activation could be post-transcriptionally mediated by the PIP5K1$\alpha$/AKT axis [120]. The AR overexpression is, indeed, directly correlated to the increase of the vascular endothelial growth factor (VEGF), thus enabling tumor angiogenesis and metastasis. Moreover, androgens activate the MMP-9, which cleavages the membrane-bound VEGF, making it bioavailable for its cognate receptor [120,121]. Thus, the androgen-induced activation of AR/PIP5K1$\alpha$/AKT signaling cascade, the release of MMP-9, and the next VEGF/VEGFR complexation are able to promote PC cells invasiveness. PIP5K1$\alpha$ and MMP-9 are both coactivators of AR, promoting its translocation into the nuclei of the cells, thus facilitating the PC growth and spreading [120,122].The somatic knockdown or the pharmacological inhibition of PIP5K1$\alpha$ block the cell cycle and impair the migratory phenotype of PC cells. The PIP5K1$\alpha$ inactivation or the deletion of its N-terminal region promote the downregulation of AR, cyclin-dependent kinase 1 (CDK1), and MMP-9. As a consequence, the tumor proliferation and invasion are blocked [123].

The androgen/AR axis plays a substantial role in the regulation and expression of MMPs and TIMPs in prostate cancer invasion and metastasis. AR could act as a transcription factor, inducing the expression of target genes, or it could encompass gene repression

at the promoter level. While in the normal rat ventral prostate, androgen represses the transcription of MMP-9, MMP-2, and their inhibitors TIMP-1 and TIMP-2 genes [124], in PC, androgen upregulates the MMP-9 expression [125] and downregulates the TIMP-2 inhibitor expression [126]. By analyzing the MMP-2 promoter, Li and colleagues [125] found two putative ARE-like motifs in the AR promoter region responsible for androgen-induced expression of MMPs genes. These motifs have been found in regions within or proximal to the promoter, or even several kilobases away upstream from the promoter [127]. These findings show that androgens can probably promote/inhibit MMPs expression through different molecular mechanisms in normal vs. pathological conditions.

In addition to the androgen-induced genomic regulation of MMP transcription, nongenomic actions can also promote the activation and release of MMPs in PC. In prostate CAFs, androgens promote the formation of the AR/FlnA/β1 integrin complex, which, in turn, recruits the membrane-type 1 protein (MT1-MMP) anchored on MB. MT1-MMP recruits pro-MMP-2 in the stroma, which is then cleaved and activated [10]. Thus, the AR/FlnA/β1 integrin complex seems to be actively involved in the formation of ECM pores that allow stromal cells to invade and surround the tumor itself. These findings show that stromal AR mediates changes in ECM composition. CAFs release the MMP-2 as inactive proenzyme (pro-MMP-2) in TME. MT-MMP-1 activates pro-MMP-2, promotes the release of MMP-2, and allows ECM degradation, as well as cell invasiveness [10,14]. Such a mechanism involves β1 integrin not only in PC [128] but also in mammary cells [129]. Again, CAFs release different factors [10] that fuel cancerous cells and facilitate their growth.

Tumor growth, spreading, and angiogenesis also depend on the increased bioavailability of signaling molecules, such as growth factors and cytokines. By liberating them from the ECM or shedding them from the cell surface, MMPs and ADAMs enable the accessibility of these factors to cancer cells and its microenvironment [130].

Consistently, the increase of MMP-9 in AR-expressing PC cells makes the membrane-bound vascular endothelial growth factor (VEGF) available for its receptor, VEGFR2 [131]. Thus, the functional link between AR and MMP-9 is critical also for ECM degradation and vascular remodeling during PC metastasis.

In addition to MMPs, many findings have shown a correlation between the male sex steroid hormones and ADAM regulation [103,132,133]. It was firstly demonstrated that ADAM proteins are regulated by androgens in PC [103]. To date, it is well accepted that different ADAMs (ADAM-9, -10, -11, -15, and -17) are expressed by PC androgen-dependent LNCaP cells and are specifically regulated (ADAM-9, -10, and -17) by androgens in PC. Dihydrotestosterone (DHT) acts on the intracellular localization of ADAM-10 [132] and upregulates ADAM-9 and -10 mRNA, while downregulating ADAM-17 mRNA [103]. By promoting the solubilization and activation of TNF-α, ADAM-9 and -10 induce the motility of androgen-stimulated PC cells. Moreover, ADAM-10 plays a role in the cleavage of MB and ECM, given its ability to degrade type-IV collagen. Through the binding with integrin αVβ3, ADAM-9, -10, and -15 can promote cell–cell and cell–ECM interactions, thus enhancing ECM remodeling and PC cells spreading [103].

Taken together, these data support the idea that androgens act at different steps of the neoplastic process. On one hand, they promote the growth and spreading of cancer cells; on the other hand, androgens regulate many functions of the surrounding stroma and ECM. In this context, specific targeting of MMPs might represent a valuable strategy to "cutting off" cell–cell communication.

## 5. Concluding Remarks

PC remains a major public health problem in Europe, and given the expected increase in the disease burden, it is a priority to rapidly translate basic research findings into novel and effective tools for patient therapy. Many advances have been made in this direction. Nevertheless, pieces of the PC puzzle are still missing, and PC frequently acquires drug resistance, often characterized by metastatic spreading. PC shows, indeed, a marked heterogeneity, characterized by rapid changes as a consequence of genetic and

epigenetic pressures. Moreover, the TME seems to play a role in the disease pathogenesis and progression. The last years have seen significant advances in understanding the molecular basis of pressures exerted by TME components on PC cells. In this intricated plot, the role of ECM and MMPs is undeniable, although still controversial. ECM components play a key role in the initiation and progression of tumors by regulating different steps of the cancer process [134], altering the TME and modulating the differentiation, migration, and infiltration, as well as polarization, of immune cells in the TME [135]. Thus, as a result, the development of an inflamed TME occurs [136]. In this context, ECM components might serve as biomarkers to improve patients' stratification, but also could be used as therapeutic targets in combination with other therapies.

In this paper, we analyzed and discussed the more recent findings on the role of MMPs released by both stromal and PC cells, with particular focus on the androgen regulation of these events. Many aspects of this topic are not well understood, although it appears that inhibiting the action of MMPs with neutralizing antibodies or specific drugs might significantly expand the arsenal of therapeutics available by urologists and oncologists. Preclinical findings have reported in the past the efficacy of MMP inhibition in several cancer models, and many synthetic MPIs have thus far been developed and tested in clinical trials, with disappointing results [137]. To date, however, only few clinical trials are ongoing in PC patients, since MPIs often need to be delivered through nanoparticles or nanomaterials. Nowadays, this strategy only represents a "proof of concept", because of the absence of reproducibility, safety, effectiveness, and scale-up production. As such, a more reliable MPIs-delivery strategy would offer great advances in the generation of new therapies for PC. Criteria to discriminate responders from nonresponders prior to the starting of the treatments are urgently needed. The discovery of predictive biomarkers, however, might allow the selection of patients who are more likely to respond. Further studies with the aim to analyze the levels of circulating MMPs together with androgens could be a valid strategic tool to find new clinical correlations.

**Author Contributions:** Conceptualization, G.C. and M.D.D.; software, R.D. and G.G.; writing—original draft preparation, C.S., R.D., M.D.D. and G.C.; writing—review and editing, P.G., B.P., M.D.D. and G.C.; visualization, C.S.; supervision, G.C. and M.D.D.; funding acquisition, G.C., A.M. and M.D.D. All authors have read and agreed to the published version of the manuscript.

**Funding:** This research was funded by Italian Ministry of University and Scientific Research (P.R.I.N. 2017EKMFTN_002 to G.C.), Regione Sicilia (Progetto di Ricerca Finalizzata RF—2019—12368937 to A.M.) and Vanvitelli Young Researcher (PATG.Rice.Base.GiovaniRicercatori2022.IDEA to M.D.D.).

**Institutional Review Board Statement:** Not applicable.

**Informed Consent Statement:** Not applicable.

**Data Availability Statement:** Not applicable.

**Conflicts of Interest:** The authors declare no conflict of interest.

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
