# Peer review of "The Androgen Regulation of Matrix Metalloproteases in Prostate Cancer and Its Related Tumor Microenvironment"

_endocrines, doi:10.3390/endocrines4020027_

Round 1

Reviewer 1 Report

This is a fine review, but some of the language could be cleaned up.  The authors address the role of Androgen Receptor in CAFs themselves, but play little attention to the critical role of stromal androgen receptor and its signaling through the production of andromedins like FGF2 and PDGF.  https://doi.org/10.1073/pnas.110524318. The authors should discuss the role of the stroma in cancer vs development and how that differs or is the same, work pioneered by Cuhna, Coffee, and Issacs.  That would be of critical interest, especially given that this may just be the coopting of normal developmental pathways during carcinogenesis.

Author Response

Reviewer 1

-This is a fine review, but some of the language could be cleaned up. 

Firstly, we thank the Referee for him/her important and valuable comments. Our changes are underlined in yellow.

-The authors address the role of Androgen Receptor in CAFs themselves but play little attention to the critical role of stromal androgen receptor and its signaling through the production of andromedins like FGF2 and PDGF.  https://doi.org/10.1073/pnas.110524318.

We agree with the Referee. We have introduced the important and undeniable role of FGF2 (and other members of the FGF family, particularly FGF-7, -9, -10) and PDGF related to stromal AR in PC development and progression at pages 3 and 4, at lines 108-122 and 140-145.

-The authors should discuss the role of the stroma in cancer vs development and how that differs or is the same, work pioneered by Cuhna, Coffee, and Issacs. That would be of critical interest, especially given that this may just be the coopting of normal developmental pathways during carcinogenesis.

We thank the referee for this hint. We have introduced in the new version of our manuscript a new paragraph titled “The role of stroma in PC development and carcinogenesis” at pages 3 and 4 in which we introduce the role of stroma and particularly of the mesenchymal urogenital sinus in prostate development and function. As required, we have introduced the new refs (35-36-37-38-39-46-47-82).

Reviewer 2 Report

1.     This manuscript has significance in prostate cancer research. However, according to the title of the manuscript, androgen mediated regulation of MMPs is in focus. This part is discussed only a small portion of the manuscript. This needs to be elaborated. Such as, the authors stated that “From these findings, it has been 296 hypothesized that the AR-induced MMP9 activation is post-transcriptionally mediated by 297 the PIP5K1α/AKT axis [94].” How is this conclusion drawn? From that cited article, this needs to be clearly described. What about the transcriptional regulation of MMPs via AR? This AR mediated regulation needs to be critically described as this is the main focus of this review.

2.     Neutrophils are also a part of the TME. That part is ignored in this manuscript what are the role of AR and MMPs in neutrophil mediated TME alteration?

3.     What is the role of MPIs to AR expression and signaling? This information is important to use MPIs as therapeutic molecule.

4.     A precise discussion about prostate cancer cell mediated AR alteration and TME mediated AR alteration and their crosstalk to alter MMPs in the tumor environment and metastasis will add more critical insight in this aspect in this review.

5.     Font of the texts mentioned inside all the figures needs to be bigger.

6.     There are some typos in line number 50, 219, and 221. Please edit those.

Author Response

  1. This manuscript has significance in prostate cancer research.

Firstly, we thank the Referee for him/her important and valuable comments. Our changes are underlined in yellow.

-However, according to the title of the manuscript, androgen mediated regulation of MMPs is in focus. This part is discussed only a small portion of the manuscript. This needs to be elaborated. Such as, the authors stated that “From these findings, it has been 296 hypothesized that the AR-induced MMP9 activation is post-transcriptionally mediated by 297 the PIP5K1α/AKT axis [94].” How is this conclusion drawn? From that cited article, this needs to be clearly described.

We thank the Referee and we agree with his/her concerns. We have better described this part in the new paragraph 4 at lines 350-359 (page 10) and 360 -362 (page 11). We have also added other details concerning the correlation between the PIP5K1alpha, AR and MMP9 expressions in the regulation of the cell cycle, angiogenesis and invasion.

-What about the transcriptional regulation of MMPs via AR? This AR mediated regulation needs to be critically described as this is the main focus of this review.

Thank you for this intriguing question. We have now detailed that the androgens/AR axis can regulate MMPs and sometimes their inhibitors, TIMPs, through genomic or non-genomic actions. Particularly in normal prostate, androgens downregulate MMPs mRNA expression, while in PC cells promote the MMP mRNA expression. Additionally, in PC stromal cells AR can interact with particular membrane-type MMPs through the integrins, thus enabling the activation and release of specific MMPs and the formation of pores in the ECM surrounding the tumor. We have discussed these findings at page 10 at lines 329-336 and at page 11 at lines 363-386.

  1. Neutrophils are also a part of the TME. That part is ignored in this manuscript what are the role of AR and MMPs in neutrophil mediated TME alteration?

We thank the Referee and we agree with his/her concerns. We have discussed the role of tumor-associated neutrophils (TANs) in the TME at page 4 and we have introduced the new refs (49-50-55-59-60-61-62-63). However, the role of MMPs released by TANs remained less clear and future investigations in this direction might allow to clarify it.

  1. What is the role of MPIs to AR expression and signaling? This information is important to use MPIs as therapeutic molecule.

We thank the reviewer for this intriguing point. Still few data are present in literature about the prognostic and diagnostic role of MMPs in PC and few studies are present in which is planned the use of MMPs inhibitors. However, we have discussed this topic at lines 333-336, 439-450.

  1. A precise discussion about prostate cancer cell mediated AR alteration and TME mediated AR alteration and their crosstalk to alter MMPs in the tumor environment and metastasis will add more critical insight in this aspect in this review.

Thank you, in accordance with this point and the previous 1st point of your revision, we have discussed more extensively the intricate correlation between AR, androgen, MMPs and TME. Our new discussion is in the new paragraph 4.

  1. Font of the texts mentioned inside all the figures needs to be bigger.

Thank you. We have modified the font in all the figures.

  1. There are some typos in line number 50, 219, and 221. Please edit those.

Thank you, we have corrected them.  

Reviewer 3 Report

the authors present a very in-depth, comprehensive and nicely structures narrative review on TME in prostate cancer, with special focus on MMP with additional emphasis on their interaction with androgen axis.

in this reviewer`s opinion there are few remarks that should be discussed by the authors:

Title - should reflect in some form the focus of the manuscript on TME and ECM - a significant part of the text is dedicated to it

1. Introduction - nice brief summarization of main terms relevant for this review, scientific basis for its importance - TME concept, CAF role in it, MMP function and its relation with androgen axis - all of these synthesized in the specific aim of the study - MMPs role in PCa TME, and their androgen axis regulation. 

2. PC facts and perspectives - nicely written chapter, but in fact an extended Introduction - leading to the feeling of little repetition - maybe it should be formatted as a sub-paragraph of Introduction?

3.  the role of CAFs in ECM remodeling - this paragraph should be formatted as its own - only 3.1.

Sections 3.2 and 3.3 should be a separate paragraph dedicated to MMPs

3.1 - nicely written comprehensive paragraph on TEM and ECM - focus on CAF, their interaction with androgen axis and cell-cell interaction on epithelial-mesenchymal level. 

Section 3.2 - one of the main paragraphs of the study discussing MMPs - comprehensive description of their biology and their role in each step of cancer metastasizing. Figure 3 is a sophisticated visualization of cancer cell and TME communication, regarding MMPs system. Significant benefit of this paragraph is the inclusion of discussion on bench to bedside translation of basic research to clinical trials

section 3.3 this is the second main paragraph in the manuscript - an in-depth discussion on androgen axis effects on MMPs, reaching the conclusion that MMPs is one of the major effector systems of androgen influence on cell communication.

Section 4 Concluding remarks - nicely outlined the current status. In this reviewer`s opinion the text on future perspectives should be broadened and more specific - it is a little bit generalized at its present form

As a whole references are appropriate and relevant to the subject, but an effort should be done for them to be updated - significant part are more than 5-years old, which is problematic on topic with such rapid development in the last few years

Author Response

The authors present a very in-depth, comprehensive and nicely structures narrative review on TME in prostate cancer, with special focus on MMP with additional emphasis on their interaction with androgen axis.

Firstly, we thank the Referee for him/her important, nice and valuable comments. Our changes are underlined in yellow.

In this reviewer`s opinion there are few remarks that should be discussed by the authors:

Title - should reflect in some form the focus of the manuscript on TME and ECM - a significant part of the text is dedicated to it.

We thank the Referee.  We completely agree. Thus, we have changed the title. The new title is: “The Androgen regulation of Matrix Metalloproteases in Prostate Cancer and its related Tumor Microenvironment”.

  1. Introduction - nice brief summarization of main terms relevant for this review, scientific basis for its importance - TME concept, CAF role in it, MMP function and its relation with androgen axis - all of these synthesized in the specific aim of the study - MMPs role in PCa TME, and their androgen axis regulation. 

Thank you. We have cleaned up the language in the new version.

  1. PC facts and perspectives - nicely written chapter, but in fact an extended Introduction - leading to the feeling of little repetition - maybe it should be formatted as a sub-paragraph of Introduction?

Thank you, following your suggestion we have eliminated some repetitions and added a new, more specific paragraph to section 2. In particular, we have added the new sub-paragraph 2.1, titled “ The role of stroma in PC development and carcinogenesis”.

  1. The role of CAFs in ECM remodeling - this paragraph should be formatted as its own - only 3.1.

Thanks, we have followed your suggestion. In the new version of our manuscript, paragraph 3.1 has been removed and we have the new paragraph 3.

Sections 3.2 and 3.3 should be a separate paragraph dedicated to MMPs

Thank you for your concern. We have modified our review. In the new version, we merged these two paragraphs in paragraph 4, titled “ 4. MMPs in ECM remodeling and PC metastatic progression: the androgens contribution”.

3.1 - nicely written comprehensive paragraph on TEM and ECM - focus on CAF, their interaction with androgen axis and cell-cell interaction on epithelial-mesenchymal level. 

Thank you for your nice comment.

Section 3.2 - one of the main paragraphs of the study discussing MMPs - comprehensive description of their biology and their role in each step of cancer metastasizing. Figure 3 is a sophisticated visualization of cancer cell and TME communication, regarding MMPs system. Significant benefit of this paragraph is the inclusion of discussion on bench to bedside translation of basic research to clinical trials

Thank you for your nice comment.

Section 3.3 this is the second main paragraph in the manuscript - an in-depth discussion on androgen axis effects on MMPs, reaching the conclusion that MMPs is one of the major effector systems of androgen influence on cell communication.

Thank you for your nice comment.

Section 4 Concluding remarks - nicely outlined the current status. In this reviewer`s opinion the text on future perspectives should be broadened and more specific - it is a little bit generalized at its present form

We agree with the reviewers. We have now enriched the conclusion section in the new version of the manuscript (see page 13).

As a whole references are appropriate and relevant to the subject, but an effort should be done for them to be updated - significant part are more than 5-years old, which is problematic on topic with such rapid development in the last few years

Thank you. We have updated the references in the new version of our review.  The refs related to the years 2018-2023 are in yellow.